# Generic and Privacy-free Synthetic Data Generation for Pretraining GANs

**Kyungjune Baek**
SIT
Yonsei University
bkjbkj12@yonsei.ac.kr

**Hyunjung Shim**[*]
Kim Jaechul Graduate School of AI
KAIST
kateshim@kaist.ac.kr

## Abstract

Transfer learning for GANs successfully improves low-shot generation performance. However, existing studies show that the pretrained model using a single benchmark dataset is not generalized to various datasets. More importantly, the pretrained model can be vulnerable to copyright or privacy risks. To resolve both issues, we propose an effective and unbiased data synthesizer, namely `Primitives-PS`, inspired by the generic characteristics of natural images. Since `Primitives-PS` only considers the generic properties of natural images, the images are free from copyright and privacy issues. In addition, the single model pretrained on our dataset can be transferred to various target datasets. Extensive analysis demonstrates that each component of our data synthesizer is effective, and provides insights on the desirable nature of the pretrained model for the transferability of GANs. For better reproducibility and implementation details we provide the source code at `https://github.com/FriedRonaldo/Primitives-PS`.

## 1 Introduction

Generative adversarial networks (GANs) [7] are a powerful generative model that can synthesize complex data by learning the implicit density distribution with adversarial training. Despite the remarkable quality, GANs require at least several thousand, mostly several hundred thousand images for training. This requirement for data collection is often infeasible in practical applications (e.g., many pictures of a treasure, endangered species, or the medical images of rare disease).

The idea of transfer learning has been recently introduced to GAN training [35, 20] for low-shot generation. Following the common practice, GAN transfer learning 1) pretrains GANs on a large-scale source dataset (e.g., FFHQ) and then 2) finetunes GANs with a relatively small target dataset. As a result, developing GANs with transfer learning clearly improves the generation quality and diversity over the models trained from scratch only with the target dataset.

However, the effectiveness highly depends on how similar the source dataset is to the target dataset. According to TransferGAN [35], transfer learning can achieve the best performance when the source shares characteristics with the target. Other than the performance issue, we argue that the current pretrained models can be vulnerable to copyright and privacy issues [40]. Even for public benchmark datasets, employing them for commercial purposes is not always permitted. For this reason, one might compose her own dataset via web crawling, but filtering out the copyrighted samples is practically difficult. Besides, unresolved copyright and privacy issues might cause legal issues [28].

Recent studies [9, 12, 5] also show that the deep generative models are vulnerable to membership inference attacks, implying that privacy issues still remains beyond the copyright issues. An adversary can reconstruct a face even without additional prior information [37]. That is, we can reveal individual

---

[*]Hyunjung Shim is a corresponding author.

NeurIPS 2022 Workshop on Synthetic Data for Empowering ML Research.

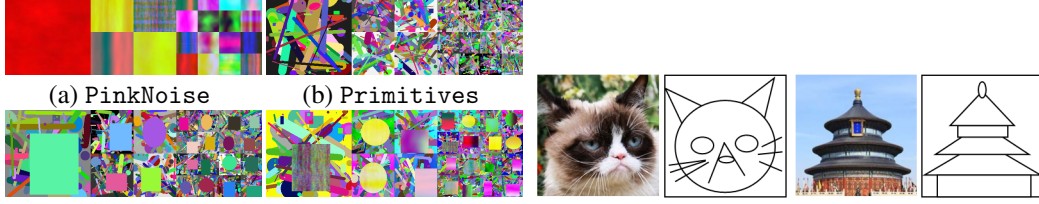

| (a) PinkNoise | (b) Primitives |
|---|---|
| (c) Primitives-S | (d) Primitives-PS |

Figure 1: Visualization of our synthetic datasets. We visualize four variants of synthetic datasets and `Primitives-PS` is finally chosen for the best performance. Images are resized in three different scales.

Figure 2: Potentials of primitive shapes for representing things. We only use a line, ellipse, and rectangle to express a cat and a temple. These examples motivate us to develop `Primitives`, which generates the data by a simple composition of the shapes.

training samples by attacking the trained model. As the network capacity of GANs increases rapidly to improve performance, the risk of memorization also grows quickly. Memorization effects make GANs more vulnerable to membership inference attacks [4]. Since we consider transfer learning, someone might argue that the membership inference on the source (e.g., pretraining) dataset should not be a critical issue. However, Zou et al. [40] reported that the membership inference of the source dataset could be conducted even after the transfer learning.

In this work, we dive into tackling the two undiscovered but critical issues of transfer learning for GANs: 1) the lack of generalization for the pretrained model and 2) the copyright or privacy issue of the pretraining dataset. Towards this goal, we adopt the generic property of the natural images in the frequency spectrum and structure. We develop our data generation strategy, namely `Primitives-PS`, inspired by the analysis and observations on natural images from previous studies [23, 30, 19]. We pretrain GANs using the synthetic dataset generated by our data synthesizer. Then, the effectiveness of the proposed method is evaluated by repurposing the pretrained model to various low-shot datasets. The experimental results verify that our synthetic dataset pretrained model achieves comparable to FFHQ pretrained model or outperforms in various low-shot datasets. We also provide the analysis on the diversity of the filters to support the result in Appendix.

## 2   Related work

**Utilizing synthetic datasets.** The samples and labels of synthetic datasets can be generated automatically and unlimitedly by a pre-defined process. Since generating synthetic data can bypass the cumbersome data crawling and pruning for data collection, previous works have utilized synthetic datasets for training the model and then achieved performance improvement on real datasets [14, 13, 27, 25, 26, 36, 31].

Although the previous methods improved the performance of the model on the real dataset, generating such synthetic datasets requires expertise in domain knowledge or a specific software (e.g., GTA-5 game engine [24]). To handle the issue, Kataoka et al. [17] utilized the iterated function system to generate fractals and used the fractals as a pretraining dataset for classification. As a concurrent work, Baradad et al. [2] observe that the unsupervised representation learning [10] trains the model using patches, and these patches are visually similar to the noise patches (from the noise generation model) or the patches drawn from GANs. However, none of the existing studies have investigated synthetic data generation for training GANs.

**Transfer learning in GANs.** GANs involve a unique architecture and a training strategy; consisting of a discriminator and generator trained via adversarial competition. Therefore, the GAN transfer learning method should be developed by considering the unique characteristics of GANs [35, 21, 34, 20, 38, 22]. TransferGAN [35] trains GANs with a small number of samples by transferring the weights trained on a relatively large dataset. FreezeD [20] fixes several layers of the discriminator and then finetunes the remaining layers. FreezeD improved the generation performance of transferring from the FFHQ pretrained model to various animals. Despite the improvement in GAN transfer learning, the model still requires a large-scale pretraining dataset. Consequently, they commonly suffer from copyright issues, and their performance is sensitive to the relationship between the source and target dataset. In contrast, our goal is to tackle both issues simultaneously by introducing an effective data synthesizer.

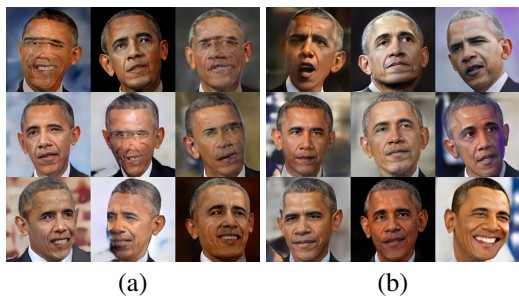

|  | (a) | | (b) |

Figure 3: Comparison between (a) `Primitives` and (b) `Primitives-PS` on Obama dataset. The model pretrained with `Primitives` generates multiple faces in a single image.

| S ╲ T | Obama | Grumpy cat | Bridge | Panda |
|---|---|---|---|---|
| Scratch + DiffAug | 48.98 | 27.51 | 57.72 | 15.82 |
| PinkNoise | 50.32 | 29.47 | 73.82 | 15.65 |
| Primitives | 43.20 | 27.97 | 59.89 | 12.78 |
| Primitives-S | 43.29 | 26.57 | 57.24 | **11.95** |
| Primitives-PS | **41.62** | **26.01** | **54.02** | 12.23 |

Table 1: The FID score of transferring to low-shot datasets from the proposed pretraining datasets. The lower is the better. Bold and underlined text indicates the best and second best performance among the pretraining datasets. It will be the same convention throughout the paper. S and T represent the source and target dataset, respectively.

**Low-shot learning in GANs.** For high-quality image generation, GANs require a large-scale dataset, and such a requirement can limit the practical use of GANs. To reduce the number of samples for training, several recent studies have introduced data augmentation for training the discriminator[33, 39, 15]. Then, the generator can produce images with a small number of samples without reflecting an unwanted transformation in the results (i.e., augmentation leakage [15]). In this work, we tackle low-shot generation using GANs via transfer learning; GANs are trained with a small number of samples by transferring a pretrained network into a low-shot dataset.

## 3 Towards an effective data synthesizer

In this work, our primary goal is to develop an unbiased and effective data synthesizer. The synthetic dataset secured by our synthesizer is then used to pretrain GANs, which facilitates low-shot data generation. To accomplish unbiased data generation, we only consider the generic properties of natural images because the inductive bias in a pretraining dataset is harmful to transfer learning of GANs. In the following, we introduce three design choices of our synthesizer inspired by the common characteristics of natural images: 1) modeling the power spectrum of images, 2) exploiting the shape primitives from natural images, and 3) adopting the existence of saliency in images. Figure 1 shows several example images by adding each aspect. For the better reproducibility and exact implementation, we provide the source code at `https://github.com/FriedRonaldo/Primitives-PS`.

### 3.1 Learning the power spectrum of natural images

Several previous works reported the magnitude of natural images in the frequency domain [6, 3, 32] roughly obeys $w_m = \frac{1}{|f_x|^a + |f_y|^a}$ where $a$ is a constant, well approximated to one. Inspired by this finding, we generate synthetic images by randomly drawing $a$ from the uniform distribution of $\mathcal{U}(0.5, 3.5)$, as also suggested in [2]. Specifically, random white noise is sampled, and then its magnitude signal after applying the Fast Fourier Transform (FFT) is weighted by $w_m$. By applying the inverse FFT to the weighted signal, we can easily compute the synthetic image. We repeat this for RGB color channels and finally produce synthetic images. Originally, the image with $a = 1$ was named a pink noise. We call this method of generating images with $a \sim \mathcal{U}(0.5, 3.5)$ as `PinkNoise`. Since we only utilize the generic properties of natural images, no inductive bias toward any specific dataset influences `PinkNoise`. As shown in Figure 1(a), `PinkNoise` produces interesting patterns with vertical, horizontal orientation, or color blobs.

## 3.2 Shape primitives inspired by natural images

*"Everything in nature is formed upon the sphere, the cone, and the cylinder. One must learn to paint these simple figures, and then one can do all that he may wish."*

Paul Cézanne

Considering the importance of phase in images (e.g., determining the unique appearance of the image [23]), `PinkNoise` alone is insufficient to represent the rich characteristics of natural images; `PinkNoise` is random noise on a phase spectrum. To have a meaningful signal even in its phase, we can consider 1) modeling the phase of natural images independently or 2) developing the different generation strategies to model the magnitude and phase simultaneously. Unlike the magnitude spectrum, we seldom find regularity in the phase of images. Besides, separately modeling the phase and magnitude may not produce meaningful images, preserving the proper structures [30]. For this reason, we focus on finding structural regularity in natural images because it can affect both magnitude and phase. Specifically, we are inspired by the observation that natural images can be represented by the composition of the elementary shapes [19]. The common practice in artistic drawings also utilizes elementary shapes as the basis for representing things (inspired by Paul Cézanne).

Figure 2 demonstrates the abstraction examples of various images using elementary shapes, such as ellipses, lines, and rectangles. We find the potential of abstraction via elementary shapes to encode the structural information of natural images and to remove the bias to a specific dataset. We then devise the data synthesizer to produce images consisting of various elementary shapes. The outputs of this synthesis procedure are akin to those of the dead leaves model [8, 18]. The dead leaves model is an early generative model, which closely mimics natural images by conducting tessellation, where their sizes and positions are determined by sampling from the Poisson process. Unlike the dead leaves model, we do not fill all the regions and use different distributions for sampling because the resultant images are quite sensitive to the hyperparameter of the Poisson process. For position, we use the uniform distribution. To prevent the large shapes in the later stage from completely overwriting those in the early stage, we gradually decrease the maximum shape size over multiple stages; drawing the small objects toward the end. In addition, it is conversely proportional to the number of currently injected shapes. We name this generation strategy `Primitives`, and Figure 1(b) visualizes the representative examples.

## 3.3 Combining saliency as prior

In addition to the natural images, we investigate the benchmark datasets and find that they commonly have saliency, target objects of interest to determine the class. These salient objects are usually located nearly in the middle of the image. For example, the animal face on the cat and panda dataset can be the saliency. To reflect the nature of benchmark datasets, we insert a large shape after applying `Primitives` and name it as `Primitives-S` (`Primitives` with Saliency).

By utilizing the three design factors, we develop four variants of our data synthesizer. They are 1) `PinkNoise` adopting the nature of magnitude spectrum of natural images only as shown in Figure 1(a), 2) `Primitives` generating various elementary (monotone) shapes randomly as illustrated in Figure 1(b), and 3) `Primitives-S` adding a salient object into `Primitives` in Figure 1(c).

Finally, we apply a `PinkNoise` pattern onto the salient object and the background of `Primitives-S`, which is called (4) `Primitives-PS` (`Primitives` with Patterned Saliency) as shown in Figure 1(d). Since the size of the salient object is considerable, having a salient monotone object may induce an unwanted texture bias. Focusing on the visual effects, inserting the monotone object can be similar to the regional dropout [29, 1] in the weakly-supervised object localization (WSOL) task. When training a network with the regional dropout, previous WSOL methods suggest filling the dropped region with mean statistics or with other regions from the same image to prevent distribution bias. Motivated by the practice in WSOL, we apply `PinkNoise` to the salient object.

The effectiveness of the proposed synthetic datasets is evaluated by transferring GANs in Section 4. We first pretrain GANs using the randomly generated images via our `Primitives-PS`, and then finetune the pretrained model on low-shot datasets. While finetuning, all competitors and our pretrained model utilize DiffAug (translation, cutout, and color jittering).

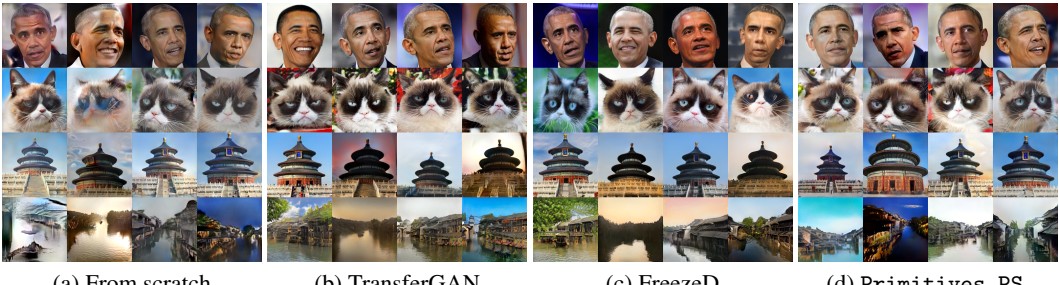

| (a) From scratch | (b) TransferGAN | (c) FreezeD | (d) `Primitives-PS` |
|---|---|---|---|

Figure 4: Qualitative evaluation on Obama, Grumpy cat, Temple, and Wuzhen.

| Source ╲ Target | Obama | Grumpy cat | Bridge | Panda | Temple | Wuzhen | Fountain | Buildings |
|---|---|---|---|---|---|---|---|---|
| Scratch + DiffAug [39] | 48.98 | 27.51 | 57.72 | 15.82 | 46.69 | 146.81 | 44.46 | 93.71 |
| TransferGAN [35] | 36.50 | 30.60 | 60.29 | 14.53 | 40.58 | 95.83 | 46.61 | 81.63 |
| FreezeD [20] | **35.90** | 29.41 | 59.47 | 13.39 | 42.09 | 93.54 | 45.70 | 80.48 |
| `Primitives-PS` | 41.62 | **26.01** | **54.02** | **12.23** | **40.42** | **88.14** | **43.06** | **78.74** |

Table 2: The FID score of transferred models to low-shot datasets. We use FFHQ pretrained weight for TransferGAN and FreezeD. For all models, we apply DiffAug.

## 4 Experiments

We first demonstrate the effectiveness of four variants of our data synthesizer. Then, we choose the best strategy among the four variants and use it for pretraining GANs. Our model is compared with other models pretrained on a natural benchmark dataset in the transfer learning scenario.

**Datasets.** For the comparison between our synthesizers, we adopt four datasets, including Obama, Grumpy cat, Panda, and Bridge of sighs (Bridge) [39]. To compare with transfer learning methods, we also use Wuzhen, Temple of heaven (Temple), and Medici fountain (Fountain). Each dataset has 100 images. In addition, we create a dataset, namely Buildings, by merging a subset of four datasets; Bridge of sighs, Wuzhen, Temple of heaven, and Medici fountain. Buildings is used to evaluate the performance under highly diverse conditions.

**Evaluation protocols.** StyleGAN2 architecture [16] with DiffAug [39] is applied when evaluating all models in the low-shot generation task. The baseline is the model trained from scratch with DiffAug. The strong competitors are TransferGAN [35] and FreezeD [20], where both methods suggest finetuning strategies. To reproduce the competitors, we used StyleGAN2 model pretrained on FFHQ and then finetune the model using TransferGAN and FreezeD, respectively. We stress that all competitors, baseline and `Primitives-PS` use DiffAug. We follow the configuration of DiffAug for `Primitives-PS` and the baseline (from scratch with DiffAug). Otherwise, we use the configuration of TransferGAN and FreezeD as described in [39] for the best performance. As an evaluation metric, we use Fréchet inception distance (FID) [11] and report the FID score of the best model during training as suggested by DiffAug [39].

### 4.1 Effects of different data synthesizers

We developed four variants of data synthesizer: `PinkNoise`, `Primitives`, `Primitives-S`, and `Primitives-PS`. We evaluate their effectiveness in the low-shot generation scenario– pretraining with the synthetic dataset and then finetuning on target datasets with DiffAug. Table 1 summarizes the FID scores of four data synthesizers and the baseline under four different low-shot datasets.

In general, `PinkNoise` fails to improve the FID score (worse than the baseline). Unlike `PinkNoise`, `Primitives` clearly improves the generation performance in Obama and Panda. However, it is not effective on Grumpy cat and Bridge. Compared to `Primitives`, `Primitives-S` further improves the FID scores, demonstrating the effectiveness of saliency prior. Finally, `Primitives-PS` clearly improves the low-shot generation performance on all datasets by about 15% on average over the baseline. From these results, we observe that 1) a naïve synthesizer (`PinkNoise`) is even worse than simply using the low-shot dataset, and 2) the combination of our three design factors (`Primitives-PS`) remarkably improves the baseline, supporting the effectiveness and importance of each factor.

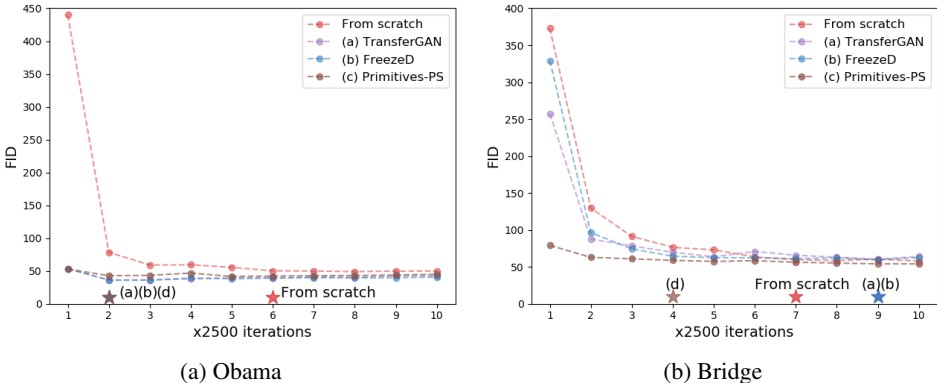

|  |  |
|:---:|:---:|
| (a) Obama | (b) Bridge |

Figure 5: FID per training iterations. The star marker (★) indicates the point where the model reaches 95% of the best FID score of the from scratch model with DiffAug (baseline).

We also verify the effect of the saliency prior by comparing `Primitives` and `Primitives-PS`. Because `Primitives` includes multiple shapes, meaning all can be candidates for the main object, the outputs often contain multiple faces in the middle of training (e.g., the top-left and the middle in Figure 3(a)). On the other hand, `Primitives-PS` focuses on generating a single face and eventually exhibits improved quality.

Considering all, we confirm that `Primitives-PS` is the best data synthesizer, and thus it is chosen as our final model for comparative evaluations with competitors.

## 4.2 Comparisons with the state-of-the-arts

We pretrain a model using `Primitives-PS` and compare it with state-of-the-art models pretrained with FFHQ in a transfer learning task to low-shot datasets.

Table 2 reports the quantitative results and Figure 4 shows the qualitative comparison. As expected, TransferGAN [35] and FreezeD [20] show outstanding performance on the Obama dataset because they are pretrained with FFHQ that is a superset of the target. Except for the Obama dataset, `Primitives-PS` pretrained model outperforms all competitors. Unless the inductive bias in the source dataset is advantageous to the target (e.g., Obama), FreezeD does not consistently outperform the baseline (from scratch with DiffAug). In fact, the performances of existing methods highly vary upon target datasets. Contrarily, our pretrained model with `Primitives-PS` consistently outperforms the competitors in each dataset, except Obama. This implies that our pretrained model has strong transferability. Since `Primitives-PS` does not use any inductive bias for modeling human faces, the performance drawback on Obama can be acceptable.

We emphasize that our achievement is impressive and meaningful in two aspects: 1) `Primitives-PS` uses no real but all synthetic images, which possesses all the attractive nature in application scenarios and 2) our results show the great potential of a single pretrained model for GAN transfer learning.

**Training convergence speed.** We investigate the convergence speed of transfer learning by examining FID upon training iterations. Figure 5 describes the evolution of the FID scores during the training. To save space, we provide two different datasets; Obama and Bridge. For Obama, all pretrained models converge faster than the baseline (from scratch with DiffAug). Meanwhile, only our model converges faster than the baseline for Bridge. Compared to the baseline, the model pretrained with `Primitives-PS` reaches 95% of the best baseline performance within the first 30% of iterations. Interestingly, other pretrained models cannot reach 95% of the best baseline performance earlier than the baseline. This shows that our model effectively reduces the required iterations for convergence, and the overhead for pretraining can be sufficiently deducted.

## 5 Limitation and conclusion

**Limitation.** Our `Primitives-PS` is devised based on the observations from natural images. Hence, it is possible that more effective observations can further improve the data generation quality. Devel-

oping a metric to quantify the transferability of the model and deriving a data generation process by optimizing the metric might be a good way to overcome this limitation.

**Conclusion.** Existing studies for GAN transfer learning utilize a model trained with natural images and thereby suffer from 1) biased pretrained model that can be harmful to the resultant performance and 2) copyright or privacy issues with both the model and dataset. To overcome these limitations, we introduce a new image synthesizer, namely `Primitives-PS`, inspired by the three generic properties of natural images: 1) following the power spectrum of natural images, 2) abstracting the image via the composition of primitive shapes (e.g., line, circle, and rectangle), and 3) having saliency in the image. Experimental comparisons and analysis show that our strategy effectively improves both the generation quality and the convergence speed.

**Acknowledgements.** This work was supported by the Korea Medical Device Development Fund grant funded by the Korea government (the Ministry of Science and ICT, the Ministry of Trade, Industry and Energy, the Ministry of Health & Welfare, Republic of Korea, the Ministry of Food and Drug Safety) (Project Number: 202011D06), the Basic Science Research Program through the National Research Foundation of Korea (NRF) funded by the MSIP (NRF-2022R1A2C3011154), IITP grant funded by the Korea government(MSIT) and KEIT grant funded by the Korea government(MOTIE) (No. 2022-0-01045) and IITP grant funded by the Korea government(MSIT) and KEIT grant funded by the Korea government(MOTIE) (No. 2022-0-00680).

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

# Appendix: Generic and Privacy-free Synthetic Data Generation for Pretraining GANs

## 1  Additional Experiments and Analysis

**Conditional generation task using CIFAR.** We conduct conditional generation via transfer learning on CIFAR-10 and 100 as summarized in Table 1. Figure 1 shows the qualitative evaluation result on CIFAR-10 with 10% of samples; our `Primitives-PS` produces the general shape and its structural components better than the baseline and DiffAug. Compared to BigGAN trained from scratch, BigGAN trained from scratch with DiffAug significantly improves the FID score, and the gain is pronounced as the number of training samples decreases. However, we observe that DiffAug suffers from augmentation leakage [1] when the samples are scarce (i.e., the generated samples contain the cutout). Our pretrained model with `Primitives-PS` shows remarkable performances under the data-hungry scenario, better than DiffAug.

However, when the samples are sufficient (100%), pretraining does not always provide gains over DiffAug. This tendency appears in various downstream tasks. Newell et. al. [2] reported that the self-supervised pretraining for semi-supervised classification is not advantageous when the amount of data-label pairs are sufficient. TransferGAN [3] showed that the gain via transfer learning decreases when the amount of samples is sufficient. In the same vein, the advantage of our pretraining with `Primitives-PS` decreases as the number of samples increases.

For the extreme low-shot scenario, we also evaluated the model trained with 1% of the dataset. Only for this evaluation, we compare three models; 1) the model naïvly trained from scratch, 2) the model trained with DiffAug only (DiffAug), and 3) our model pretrained with `Primitives-PS` and then finetuned without DiffAug. The FID score of the baseline, DiffAug, and ours are 112.13, 101.91, and 78.48, respectively. Although DiffAug improved FID, we observe that DiffAug suffers from the augmentation leakage issue. Therefore, the improvement in FID and its generation results are not meaningful. In contrast, our pretrained model can significantly improve the generation performance without any issue.

**Diverse filters matter for transferring GANs.** From the superior performances of our pretrained model, we conjecture that our achievement was possible by the unbiased nature of our dataset; the pretrained model with FFHQ (FreezeD) has an inductive bias as the face dataset. A previous study analyzing the transferability of CNN [4] also pointed out that the performance of the target dataset degrades when the filters are highly specialized to the source dataset. To analyze the transferability empirically, we measure the similarity between the filters of each layer of the pretrained model. We regard that highly diverse (less similar to each other) filters can indicate that the model is less biased towards a particular domain. That means that the highly transferable model tends to have low filter similarity on average. Specifically, given a weight matrix of each layer, its shape is $[O, I, H, W]$, where $O$ filters have $I \times H \times W$ tensors. Then, we measure the cosine similarity among all possible permutations of $O$ filters and report the average similarity of all layers in Table 3.

In summary, `Primitives-PS` shows the more diverse filter set in 21 out of 26 layers than the FFHQ pretrained model. According to [4], the higher layer (close to the output) tends to specialize in the trained dataset. The same observation holds in our discriminator. The similarity in the last layer of the FFHQ pretrained model is approximately four times higher than `Primitives-PS`. This explains

NeurIPS 2022 Workshop on Synthetic Data for Empowering ML Research.

| | CIFAR-10 | | | CIFAR-100 | | |
|---|---|---|---|---|---|---|
| | 10% | 20% | 100% | 10% | 20% | 100% |
| BigGAN | 44.14 | 20.80 | 9.45 | 66.21 | 34.78 | 13.45 |
| + DiffAug | 29.78* | 14.04 | **8.55** | 41.70* | 21.14 | 11.51 |
| + Pretrained (PS) | **21.33** | **12.79** | 8.79 | **32.57** | **20.58** | **11.29** |

Table 1: The FID of BigGAN, with DiffAug, and with DiffAug initialized by `Primitives-PS` (PS) pretrained model on CIFAR. '*' indicates the best FID before augmentation leakage [1].

| Policy | Obama | Grumpy cat | Bridge | Panda |
|---|---|---|---|---|
| **Fix** ($1/10$) | 48.30 | 29.74 | 63.00 | 17.69 |
| **Fix** ($1/5$) | 46.41 | 29.22 | 64.02 | 14.97 |
| **Fix** ($1/2$) | 48.05 | 29.37 | 64.65 | 15.14 |
| `PinkNoise + PS` | 49.13 | 29.87 | 66.00 | 15.12 |
| **Rand** | 44.85 | 29.84 | 60.45 | 14.67 |
| **Decay** | **41.62** | **26.01** | **54.02** | **12.23** |
| # of particles | Obama | Grumpy cat | Bridge | Panda |
| 0 | 49.13 | 29.87 | 66.00 | 15.12 |
| 10 | 44.10 | 28.00 | 63.26 | 13.35 |
| 50 | 42.49 | 28.40 | 59.17 | **11.79** |
| 100 | **41.62** | **26.01** | 54.02 | 12.23 |
| 500 | 42.45 | 27.92 | **52.27** | 12.12 |

Table 2: The average consine similarity between the filters in the same layer. The lower value indicates the more diverse filters.

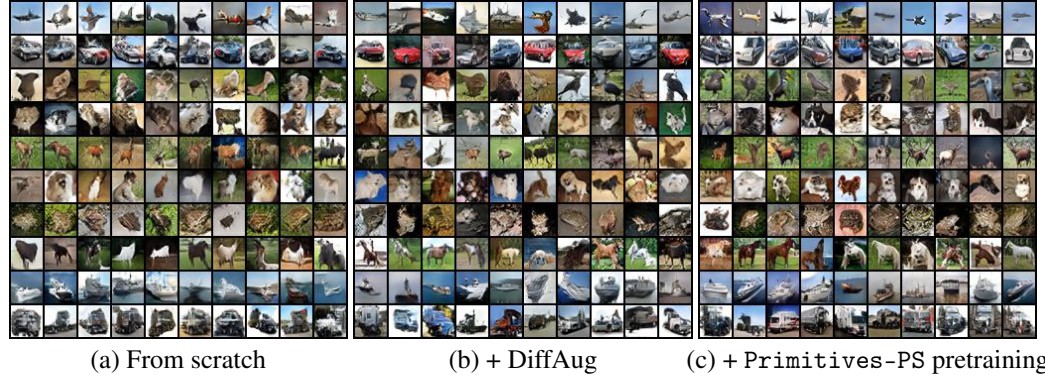

(a) From scratch      (b) + DiffAug      (c) + `Primitives-PS` pretraining

Figure 1: Qualitative evaluation on CIFAR-10 dataset with 10% of samples. Each row contains samples in the same class.

that the FFHQ pretrained model specialized in human faces, thus transferring well to Obama but not to others.

**Ablation study.** When developing `Primitives-PS`, we introduce two hyperparameters; 1) the total number of shapes and 2) the policy to determine the size of each component. For determining the size, we consider three policies; **Fix**, **Rand** and **Decay**. **Fix** indicates that all particles have the same size. To examine the effect of various scale, we set this size as $H \cdot [1/10, 1/5, 1/2]$, where $H$ is the image resolution. **Rand** randomly samples the size from the uniform distribution. Both policies can induce the occlusion of the previously injected shapes by the later shape. **Decay** can bypass the occlusion issue effectively. **Decay** arbitrarily samples the size from the uniform distribution, where the maximum size is limited to $(H \cdot 1/5 \cdot (N - n)/N)$, and $N$ and $n$ are the total number of shapes and the number of previously injected particles. In this way, we can ensure that the shapes inserted in the early stage are still visible in the final data. The upper-side of Table 2 summarizes the FID score for each policy on four datasets. The differences in FID among **Fix** policies are trivial in that their ratios are not highly correlated with their ranks. Also, we observe that the shapes at the final stage overwrite the previous shapes. Then, the overall appearance with **Fix** are similar to `PinkNoise` with a salient object. We investigate the synthesizer that combines `PinkNoise` with PS by injecting a saliency and then applying `PinkNoise` on it. Interestingly, we observe that it shows the similar FID scores to **Fix**. For **Rand**, it improves the FID score on Obama and bridge, however, the overall performance is much worse than **Decay**. Therefore, we choose a **Decay** policy as default for choosing the size.

Besides, the total number of shapes is important because it affects the transferability and the time complexity of the synthesizer. The lower-side of Table 2 demonstrates the performance trends upon the total number of shapes. A zero particle case implies that only one background and one salient object, thus equivalent to `PinkNoise + PS`. As the number of shapes ($N$) grows upon roughly 100, the performance tends to improve. However, over $N = 100$, we do not observe the consistent gain. From the ablation study, we decide $N = 100$ in each image to enjoy the reasonable performance gain and to reduce the time complexity.

|       | Discriminator |        | Generator |        |
|-------|---------------|--------|-----------|--------|
|       | Primitives-PS | FFHQ   | Primitives-PS | FFHQ |
| conv0 | **0.00660**   | 0.01245 | **0.00315** | 0.00685 |
| conv1 | 0.02104       | **0.00932** | **0.00273** | 0.00843 |
| conv2 | 0.01012       | **0.00779** | **0.00291** | 0.00956 |
| conv3 | **0.00839**   | 0.01216 | **0.00348** | 0.01080 |
| conv4 | **0.00607**   | 0.00713 | **0.00539** | 0.01059 |
| conv5 | **0.00596**   | 0.00668 | **0.00329** | 0.01406 |
| conv6 | **0.00507**   | 0.00563 | **0.00363** | 0.01199 |
| conv7 | **0.00632**   | 0.00714 | **0.00433** | 0.01465 |
| conv8 | 0.00380       | **0.00365** | **0.00652** | 0.01317 |
| conv9 | **0.00521**   | 0.00703 | **0.00933** | 0.01626 |
| conv10 | 0.00503      | **0.00420** | **0.01133** | 0.01778 |
| conv11 | **0.00462**  | 0.00760 | 0.01981 | **0.01977** |
| conv12 | **0.01844**  | 0.08438 | **0.03176** | 0.03250 |
| Mean  | **0.00820**   | 0.01348 | **0.00828** | 0.01434 |

Table 3: Ablation study on the policy to determine the size of each particle (upper) and the number of particles (lower).

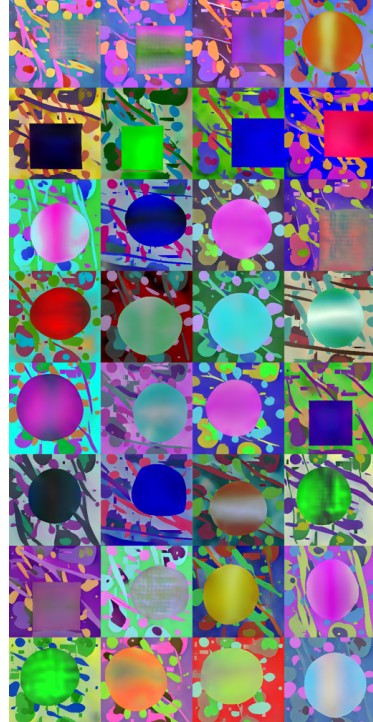

Figure 2: The outputs of the model pretrained with `Primitives-PS`. The generated outputs are similar to the synthetic samples.

## 2 Pretraining Results and Details

We provide the outputs of the generator pretrained with `Primitives-PS`. For pretraining, we train the model during 800K images with batch size = 16, therefore, the total number of iterations is 50K. For finetuing all the models, we train the model during 400K images. The generated (fake) synthetic images are similar to the real synthetic samples as shown in Figure 1 of the main text.

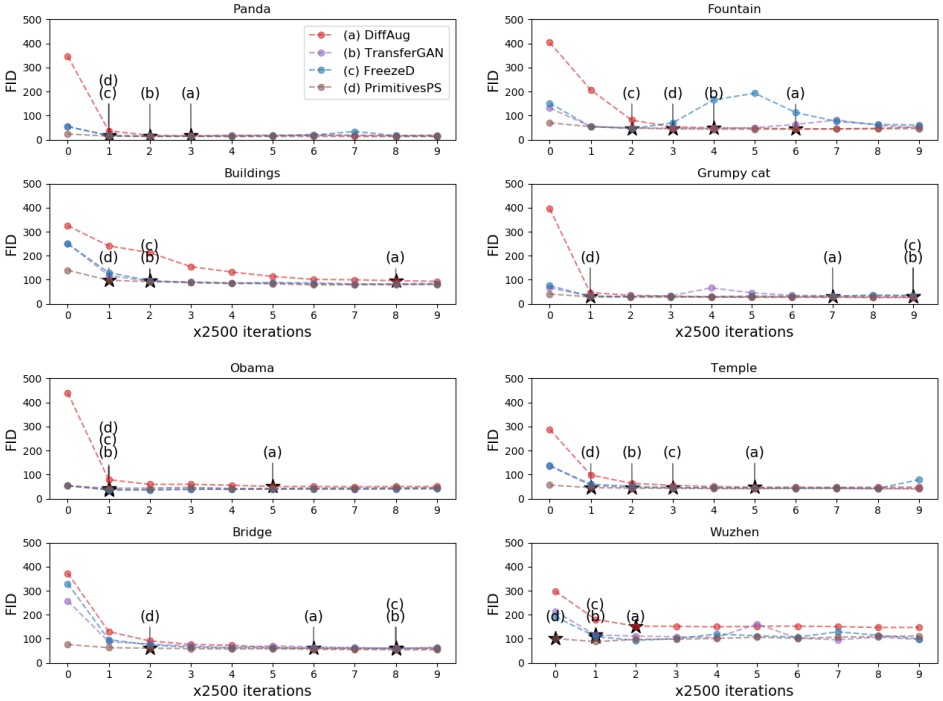

Figure 3: The additional results of Figure 6 in the main text. FID per training iterations. The star marker (★) indicates the point where the model reaches 95% of the best FID score of the from scratch model with DiffAug (baseline). The legend is the same for all graphs.

## 3 Convergence Speed of Transfer Learning Methods

Figure 3 shows the evolution of the FID scores during the training of the transfer learning methods. The model pretrained with our synthetic dataset exhibits comparable or faster convergence than the competitors that are pretrained on FFHQ. Herein, we observe the convergence speed in terms of the number of iterations to reach 95% of the best FID score of the baseline (from scratch model with DiffAug).

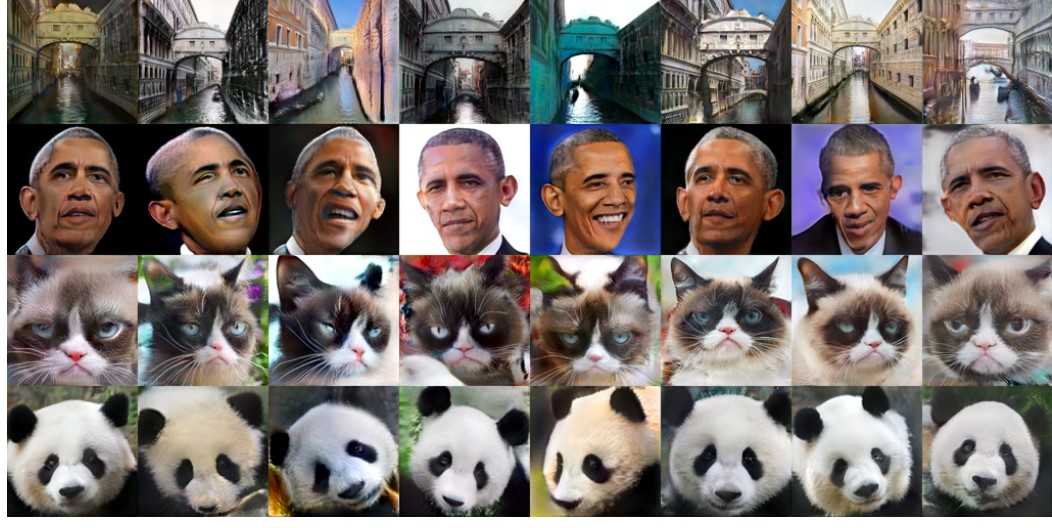

(a) `PinkNoise`

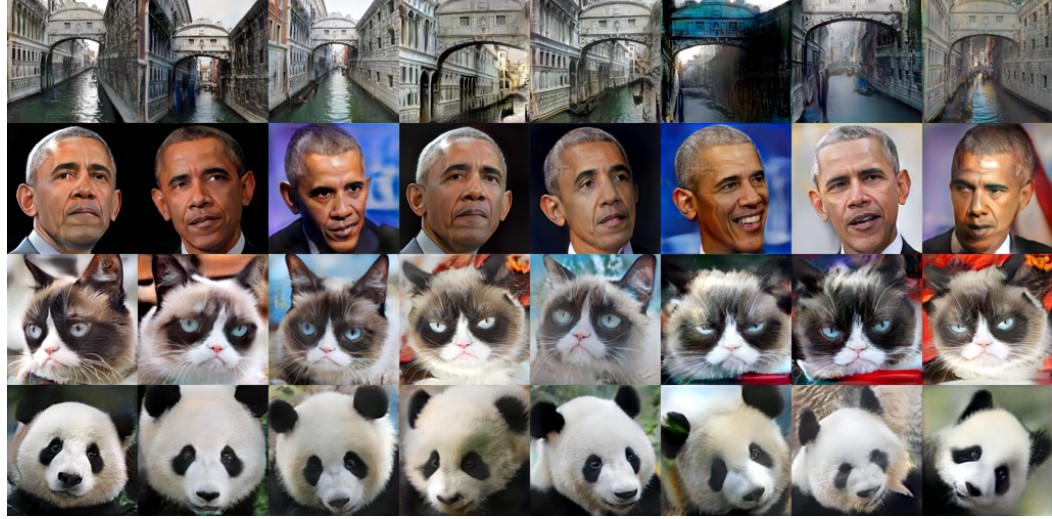

(b) `Primitives`

Figure 4: Low-shot image generation results of the models transferred from `PinkNoise` and `Primitives`.

# 4 Qualitative Comparison Among Data Synthesizers

In addition to the quantitative comparison of our data synthesizers, we also qualitatively compare our four variants of the data synthesizer used for quantitative evaluation. From the first to the last row, Bridge of sighs, Obama, Grumpy cat, and Panda. `PinkNoise` generates the images with unstructured samples (e.g. Obama and Grumpy cat) and the outputs of `Primitives` on Panda have lower fidelity (e.g. the last three samples). Compared to `PinkNoise` and `Primitives`, `Primitives-S` and `Primitives-PS` provide plausible samples. Between the last two synthetic datasets, `Primitives-S` sometimes drops the important factor, for example, the eyes of the cat (6-th column). While `Primitives-PS` generates more diverse and plausible samples than the other synthetic datasets.

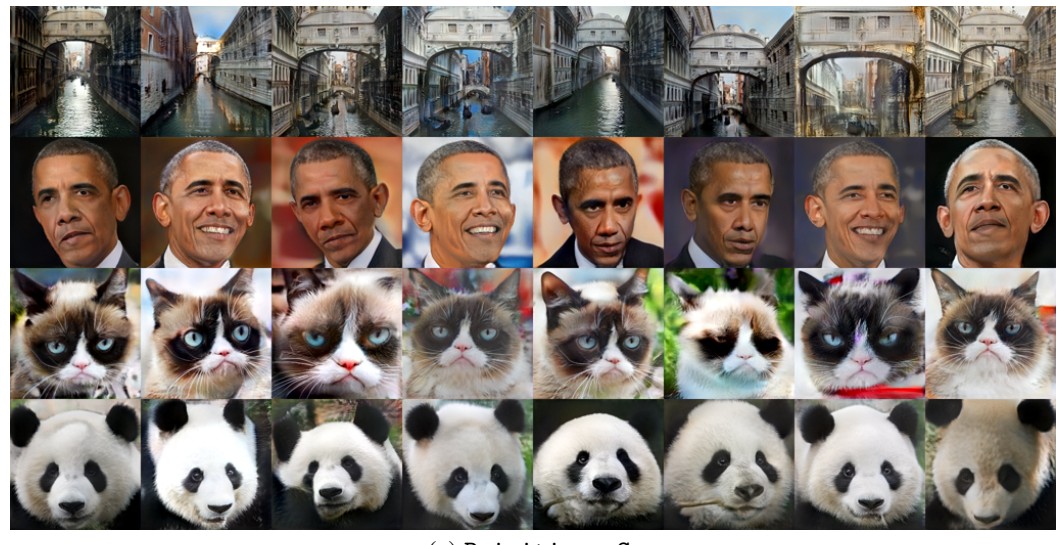

(a) `Primitives-S`

(b) `Primitives-PS`

Figure 5: Low-shot image generation results of the models transferred from `Primitives-S` and `Primitives-PS`.

## 5  Qualitative Comparisons With Competing Transfer Learning Methods

In addition to the quantitative comparison, we also provide the qualitative comparisons on eight datasets that are used for quantitative evaluation in the main text. From the first to the last row, Buildings, Bridge of sighs, Obama, Medici fountain, Grumpy cat, Temple of heaven, Panda, and Wuzhen. In terms of fidelity of the generated images, our `Primitives-PS` outperforms the competitors. Especially, Grumpy cat images generated by the competitors often do not contain eyes or have only part of the face. Because of the size, we show one figure per page.

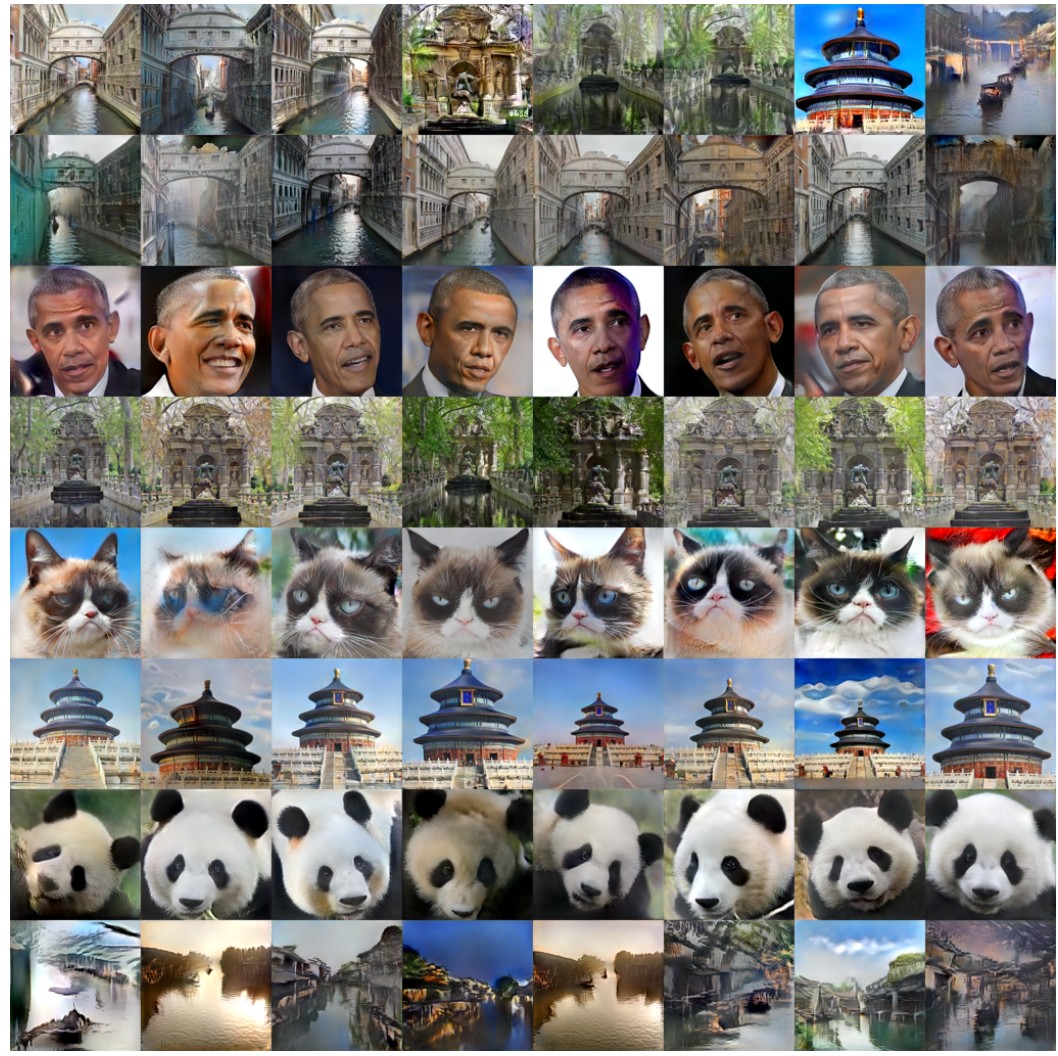

Figure 6: The additional generated samples of Figure 5 in the main text. The images are generated with the model trained from scratch.

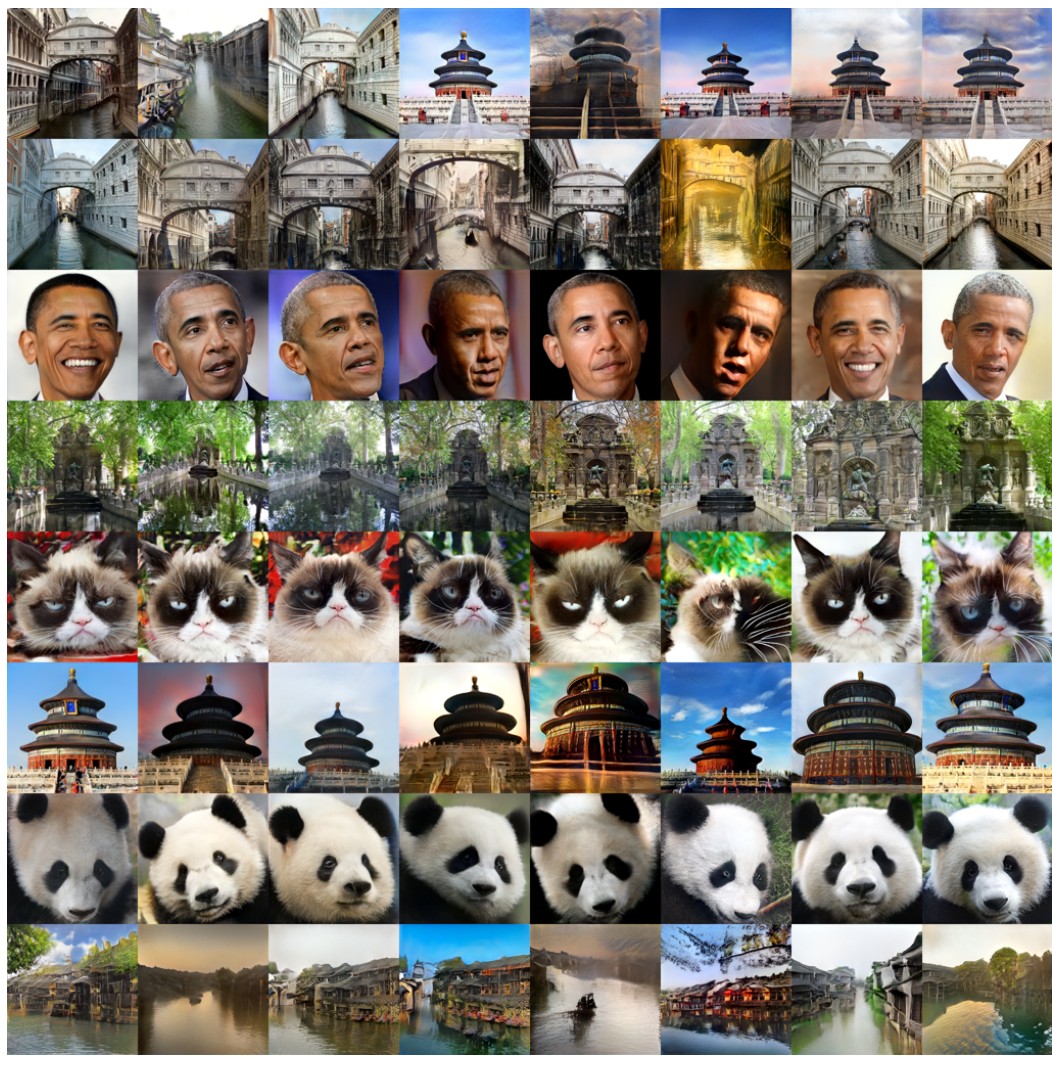

Figure 7: The additional generated samples of Figure 5 in the main text. The images are generated with the model pretrained with FFHQ and transferred by using TransferGAN.

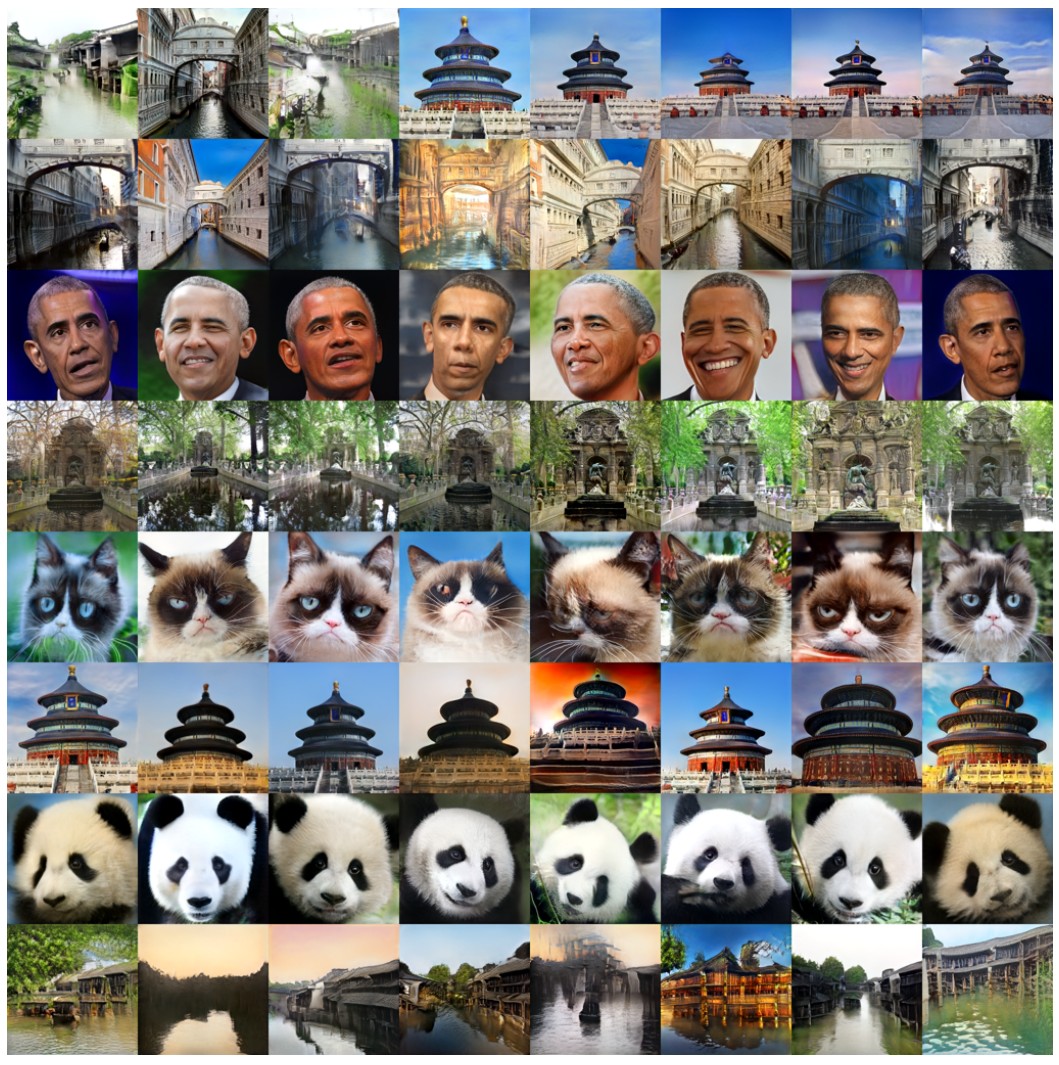

Figure 8: The additional generated samples of Figure 5 in the main text. The images are generated with the model pretrained with FFHQ and transferred by using FreezeD.

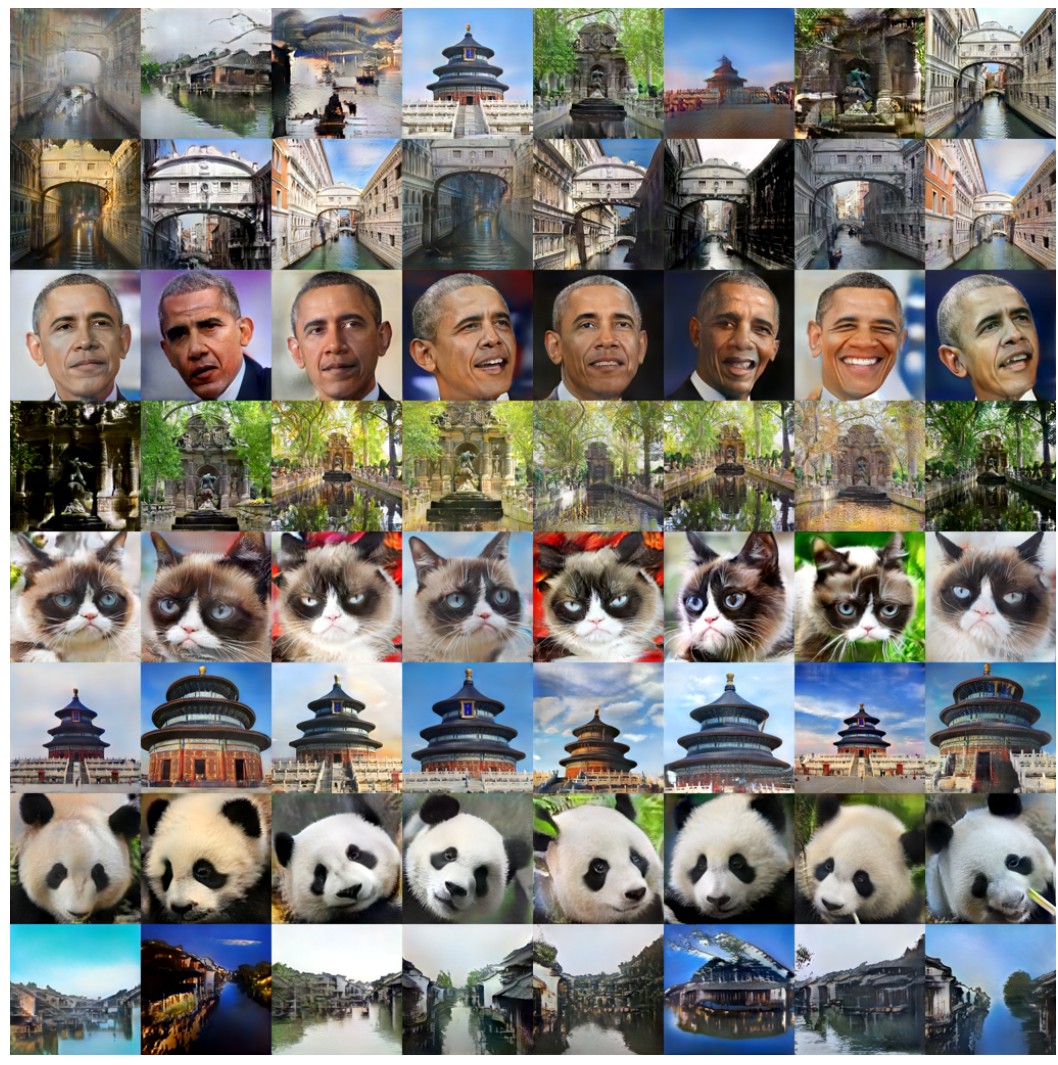

Figure 9: The additional generated samples of Figure 5 in the main text. The images are generated with the model pretrained with our `Primitives-PS`.

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
