# OpenReview forum: "Generic and Privacy-free Synthetic Data Generation for Pretraining GANs"
_NeurIPS.cc/2022/Workshop/SyntheticData4ML — Neurips 2022 SyntheticData4ML_

### Official Review · Reviewer_ymPc · 2022-10-07
**Review paper #8 - strong accept**

**Rating:** 9
**Confidence:** 5

**Review:**

Title: Generic and Privacy-free Synthetic Data Generation for Pretraining GANs (Paper #8)

Subject matter:
1. The paper uses transfer learning to ease the workload of GAN training (“pre-training”) in the context of image synthetiation;
2. Pretrained GANs only work when the data used in pretraining shares characteristics with the target synthetic data the GAN should produce; furthermore, pretrained models may incur copyright issues and risk of membership inference attacks;
3. The authors propose a pretraining method for GANs that overcome these obstacles through initializing GANs with generalized patterns.
4. Four methods are designed with increasing complexity: 1) color and magnitude generation through scaling random samples (“pink noise”); 2) Pink noise with structural information generation in the form of elementary geometric shapes (“primitives”); 3) Addition of saliency to primitives by locating crucial (“salient”) shapes at relevant places (“primitives-s”) and 4) Adding pink noise to the salient object in primitive-S (“primitive-ps”);
6. Of these four methods, primitive-ps is selected as the best in experiments;
7. Further experiments are then conducted to compare primitive-ps to GANs starting with random samples and two pretrained GAN models; results indicate that primitive-ps outperforms all other methods in seven of eight cases.

Relevance:
1. The topic is highly relevant to the scope of the workshop;
2. The methodology is novel and explained with clarity;
3. The results indicate that the method is suitable as a starting point for GANs, significantly outperforming starting from random sampling or starting from pretrained models;
4. Combined, these points make the paper highly relevant and novel.

Language, structure and formatting
1. The paper is very well-written; the language is clear and the overall structure is well-organized;
2. Scale differences in the initial iterations make Figure 5 hard to interpret;
3. The paper uses many references, which helps place it in context. Some of these actually distract a bit from the core message, most notably the references to the dead leaves model. This seems to fit better in a “prior research” section than in a section on methodology. This is, however, somewhat subjective and understandable given the space requirements.
4. (Not an important point) I wonder if the moniker “pretraining” technically applies to the method.

---

### Official Review · Reviewer_Lu16 · 2022-10-18
**Well written paper with intuitive approach for improving few-shop generation in GANs**

**Rating:** 7
**Confidence:** 4

**Review:**

The paper discusses the training of transfer learning generative models in the few-shot scenario. The authors propose pre-training GANs with synthetic data and provide comparisons to other transfer learning-based approaches.

Pros:

1. The paper is well written and well organized.
2. The contributions of the paper are clear.
3. The experiments are thorough - providing both qualitative assessment (generated images) and quantitative assessment (FID scores).
4. The proposed pre-training algorithm generalizes better (across different image datasets) .

Cons:

1. Tables 1 and 2 show FID scores of the best performing model for each of the datasets, yet the number of models trained per dataset was not specified.
2. Privacy was described as a motivation, yet no experiments highlighting the benefits of the approach (e.g., against membership inference attacks) were demonstrated.


Suggestions:

1. When reporting results, instead of using the best performing model - show the average FID across different models and report the standard error. This will give readers an intuition on the expected performance of the pre-training approaches, as well as the variation across different random seeds.
2. Provide a more clear explanation of Section 3.1. Several notations are not defined (e.g., $f_x$ and $f_y$).

---

### Official Review · Reviewer_fQuc · 2022-10-19
**Good paper overall, needs to provide more technical details**

**Rating:** 6
**Confidence:** 4

**Review:**

The paper tackles an important problem of pretraining GAN for image generation by a single benchmark dataset that can be easily generalizable to multiple image datasets (including those datasets pretraining for which could not be easily done in advance due to the dataset privacy concerns). Once the GAN is pretrained - the idea is to tune it to the dataset of interest fast and by using a small number samples.

The problem and importance of the task in consideration are well described and motivated. Overall, the structure of the paper is reasonably clear and the proposed transfer experiments and outcomes are well sound.

The paper needs to provide more technical details on the construction of Primitives-PS. Particularly, the descriptions in sections 3.1, 3.2, 3.3 should be better elaborated as they lie in the gist of Primitives-PS construction. Section 3.2 (learning the shape of natural images) is particularly vague and needs to be explained better. Where appropriate mathematical justifications/schematics of architectures used need to be provided. Progressive differences (or ablation study) between PinkNoise, Primitives, Primitives-S, Primitives-PS should be presented on a more technical/visual level.

Also, it's worth elaborating on cases when Primitives-PS approach should not be objectively expected to achieve high outcomes.

---

### Meta-Review · Area_Chair_s7Bx · 2022-10-20

**Recommendation:** Accept